

# Urine proteomic analysis of the rat e-cigarette model

Yuqing Liu, Ziyun Shen, Chenyang Zhao and Youhe Gao

Gene Engineering Drug and Biotechnology Beijing Key Laboratory, Beijing Normal University, Beijing, China

## ABSTRACT

**Background.** We were curious if the urinary proteome could reflect the effects of e-cigarettes on the organism.

**Methods.** Urine samples were collected from a rat e-cigarette model before, during, and after two weeks of e-cigarette smoking. Urine proteomes before and after smoking of each rat were compared individually, while the control group was set up to rule out differences caused by rat growth and development.

**Results.** Fetuin-B, a biomarker of chronic obstructive pulmonary disease (COPD), and annexin A2, which is recognized as a multiple tumour marker, were identified as differential proteins in five out of six smoking rats on day 3. To our surprise, odourant-binding proteins expressed in the olfactory epithelium were also found and were significantly upregulated. Pathways enriched by the differential proteins include the apelin signalling pathway, folate biosynthesis pathway, arachidonic acid metabolism, chemical carcinogenesis-DNA adducts and chemical carcinogenesis-reactive oxygen species. They have been reported to be associated with immune system, cardiovascular system, respiratory system, *etc.*

**Conclusions.** Urinary proteome could reflect the effects of e-cigarettes in rats.

## INTRODUCTION

### E-cigarettes

E-cigarettes are mainly composed of four parts: soot oil, a heating system, a power supply and a filter nozzle. Aerosols with specific odours are generated by heating atomization for smokers. The main components of the aerosol liquid of the electronic vapourizer are plant glycerin, propylene glycol, edible flavour and nicotine salt. As of 2019, approximately 10 million people aged 15 years and older in China had used e-cigarettes (*Xinhuanet, 2019*). The population using e-cigarettes is predominantly young adults, with the highest use in the 15- to 24-year age group (*Xinhuanet, 2019*). The vast majority (58.3%) of middle school e-cigarette users use fruit-flavoured e-cigarettes, and previous research suggests that these tastes may attract young people to try e-cigarettes (*Bold et al., 2016*). On May 1, 2022, the *Regulations on the Administration of Electronic Tobacco* prohibited the use of electronic cigarettes with flavours other than tobacco tastes (*State Council Gazette, 2022*). *Pipe & Mir (2022)* found that the composition of heated chemical aerosols inhaled by the human body after electron fumigation is very complex and includes nicotine, nitrosamines, carbonyl compounds, heavy metals, free radicals, reactive oxygen species,

Corresponding author
Youhe Gao, gaoyouhe@bnu.edu.cn

particulate matter, and "emerging chemicals of concern", which further demonstrates the potential harm of smoking e-cigarettes. Studies have shown that smoking e-cigarettes may increase the risk of lung disease (*Cheng et al., 2022*) and cardiovascular disease (*Fetterman et al., 2020*; *Lee et al., 2019*) and may cause harm to the liver (*Espinoza-Derout et al., 2019*), urinary system (*Lee et al., 2018*), and immune system (*Martin et al., 2016*). In pregnant women, exposure to e-cigarettes causes harm to the mother and the foetus. *Ballbè et al. (2023)* detected low but nonnegligible concentrations of e-smoke-associated analytes in cord blood and breast milk of nonuser pregnant women exposed to e-cigarettes. *Aslaner et al. (2022)* also demonstrated that the inhalation of secondhand e-cigarette smoke by pregnant women can have long-term effects on the lungs of offspring. At the same time, because the nicotine content in e-cigarette smoke is equivalent to or even higher than that in combustible smoke (*Ballbè et al., 2014*), the damage caused by e-cigarette smoke to the human body cannot be ignored.

## Urine biomarkers

Biomarkers are indicators that can objectively reflect normal pathological processes as well as physiological processes (*Strimbu & Tavel, 2010*), and clinically, biomarkers can predict, monitor, and diagnose multifactorial diseases at different stages (*Gerszten Robert & Wang Thomas, 2008*). The potential of urinary biomarkers has not been fully explored compared to the more widely used blood biomarkers, especially in terms of early diagnosis of disease and status prediction. Because the homeostatic mechanisms are regulated in the blood, changes in the blood proteome caused by a particular disease are metabolically excreted, and no significant changes are apparent in the early stages of a disease. In contrast, as a filtrate of the blood, urine bears no need or mechanism for stability; thus, minor changes in the disease at an early stage can be observed in urine (*Gao, 2013*), which indicates that urine is a good source of biomarkers.

Currently, the detection of biomarkers in urine has attracted increasing attention from clinicians and researchers. This approach has been used in the treatment and research of a variety of diseases, such as pulmonary fibrosis (*Wu et al., 2017*), colitis (*Qin et al., 2019*), glioma (*Wu et al., 2020*) and other diseases. Studies have shown that urine biomarkers can classify diseases, such as predicting chronic kidney disease (CKD) (*Hao et al., 2020*) and distinguishing benign and malignant ovarian cancer (*Ni et al., 2021*). Urine biomarkers can also be used to detect whether complete resection and recurrence occur after tumour surgery so that adjustments can be made in time to reduce the risk of recurrence. Pharmacologically, urinary biomarkers may indicate the utility of drugs in the body, such as predicting the efficacy of rituximab therapy in adult patients with systemic lupus erythematosus (SLE) and determining that sacubitril-valsartan is more effective than valsartan in the treatment of chronic heart failure (*Davies et al., 2021*). In terms of exercise physiology, urine biomarkers can reflect changes in urine proteomics after exercise, thus providing a scientific basis for the rational training of athletes (*Meng et al., 2021*). In recent years, many studies have shown that urine proteomics can also reveal biomarkers in neurodegenerative diseases and psychiatric diseases, such as Parkinson's disease (*Virreira Winter et al., 2021*; *Seol, Kim*

*& Son, 2020*), Alzheimer's disease (*Watanabe et al., 2020*), depression (*Huan et al., 2021*), autism (*Meng, Huan & Gao, 2021*) and other diseases.

However, there have been no studies on e-cigarettes in the field of urine proteomics. The urine proteome is susceptible to multiple factors, such as diet, drug therapy, and daily activities. To make the experimental results more accurate, it is critical to use a simple and controllable system. Because the genetic and environmental factors associated with animal models can be artificially controlled and the influence of unrelated factors can be minimized, the use of animal models is a very appropriate experimental method. Therefore, we constructed an animal model to analyse the urine proteomics of the rat e-cigarette model, and the experimental workflow is shown in Fig. 1. We aimed to determine the effect of smoking e-cigarettes on the urine proteome of rats.

# MATERIALS AND METHODS

## Rat model establishment

Portions of this text were previously published as part of a preprint (*Liu et al., 2022*). Eleven SPF (specific pathogen free) 8-week-old healthy male Wistar rats (180–200 g) were purchased from Beijing Vital River Laboratory Animal Technology Co., Ltd. (Beijing, China), with the animal licence number SYXK (Jing) 2021-0011. All rats were maintained in a standard environment (room temperature ($22 \pm 1$) °C, humidity 65%–70%). The environmental equipment for animal experiments met the requirements of the standards for experimental animal grades, and qualified feed, cages, bedding and other supplies were used. All rats were kept in a new environment for three days before starting the experiment. At the end of the experiment, the rats were euthanized according to the standard. Our method of euthanasia was cervical dislocation following anaesthesia. Anaesthesia was induced with 0.41 mL of 2% isoflurane per minute. All experimental procedures were reviewed and approved by the Ethics Committee of the College of Life Sciences, Beijing Normal University (Approval No. CLS-AWEC-B-2022-003). All experimental procedures were carried out and reported in compliance with the ARRIVE guidelines.

The animal model of e-cigarette use was established by randomly dividing 11 rats into an experimental group and a control group. Five of the control rats were maintained in a standard environment for 17 days. Six rats in the experimental group smoked e-cigarettes once a day during the same period. We used a syringe to simulate the process of inhaling e-cigarette smoke from a human mouth and injected the resulting smoke into the rat cages. One-third of the 3% nicotine e-cigarette refill cartridges was made into smoke (approximately 16 mg of nicotine) and evenly injected into two cages (36 cm (length) × 20 cm (width) × 28 cm (height)). Under the condition of ensuring adequate oxygen content, three rats in the experimental group were placed in each cage and continued to smoke for 1 h for 14 days. They were returned to their original cages after each smoking treatment. Rats were observed for behavioural changes during the experiment, and body weights were recorded every 5 days.

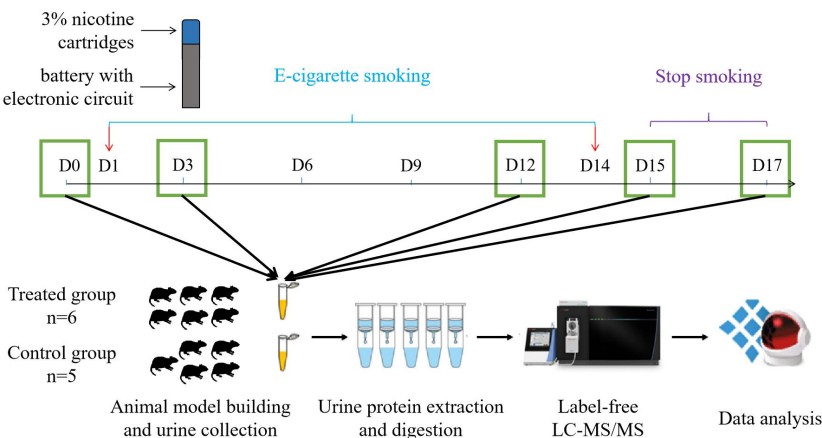

**Figure 1 Workflow for urine proteomic analysis in rat e-cigarette models.** In the experimental group, urine samples were collected before and on days 3, 6, 9, and 12 after smoking e-cigarettes and on days 1 and 3 after stopping smoking e-cigarettes. After urine samples were collected and processed in the experimental and control groups, the protein groups of the two groups were identified using liquid chromatography coupled with tandem mass spectrometry (LC–MS/MS) to quantitatively analyze the damage caused to the rat body at different stages of smoking e-cigarettes.

## Urine collection

After all rats were kept in a new environment for three days, they were uniformly placed in metabolic cages to collect urine samples for 12 h. Urine samples were collected during a 12-h period in metabolic cages (DXL-D, made by Beijing Jiayuan Xingye Technology Co., Ltd.) from all rats on days 3, 6, 9, and 12 of e-cigarette smoking and on days 1 (equivalent to day 15) and 3 (equivalent to day 17) of cessation of e-cigarette smoking. Rats were fasted and water-deprived during urine collection, and all collected urine samples were stored in a −80 °C freezer.

## Treatment of the urine samples

We wanted to observe the sensitivity of the urine proteome and to observe whether the changes within the rat body could be reflected in the urine proteome after smoking e-cigarettes for a short time. Therefore, five time points, namely, nonsmoking day 0, smoking days 3 and 12, and smoking cessation days 1 and 3, were selected as the samples for this focused analysis.

Urine protein extraction and quantification: Rat urine samples collected at five time points were centrifuged at 12,000×g for 40 min at 4 °C, and the supernatants were transferred to new Eppendorf (EP) tubes. Three volumes of precooled absolute ethanol was added, and the samples were homogeneously mixed and precipitated overnight at −20 °C. The following day, the mixture was centrifuged at 12,000×g for 30 min at 4 °C, and the supernatant was discarded. The protein pellet was resuspended in lysis solution (containing 8 mol/L urea, 2 mol/L thiourea, 25 mmol/L dithiothreitol, and 50 mmol/L Tris). The samples were centrifuged at 12,000×g for 30 min at 4 °C, and the supernatant

was placed in a new EP tube. The protein concentration was measured by the Bradford assay.

Urine proteins were digested with trypsin (Trypsin Gold, Mass Spec Grade, Promega, Fitchburg, Wisconsin, USA) using FASP methods (*Wisniewski et al., 2009*). Urinary protease cleavage: A 100-µg urine protein sample was added to the filter membrane (Pall, Port Washington, NY, USA) of a 10-kDa ultrafiltration tube and placed in an EP tube, and 25 mmol/L $NH_4HCO_3$ solution was added to make a total volume of 200 µL. Then, 20 mM dithiothreitol solution (dithiothreitol, DTT, Sigma, St. Louis, MO, USA) was added, and after vortex mixing, the metal bath was heated at 97 °C for 5 min and cooled to room temperature. Iodoacetamide (Iodoacetamide, IAA, Sigma, St. Louis, MO, USA) was added at 50 mM, mixed well and allowed to react for 40 min at room temperature in the dark. Then, the following steps were performed: ① membrane washing –200 µL of UA solution (8 mol/L urea, 0.1 mol/L Tris–HCl, pH 8.5) was added and centrifuged twice at 14,000×g for 5 min at 18 °C; ② Loading –freshly treated samples were added and centrifuged at 14,000×g for 40 min at 18 °C; ③ 200 µL of UA solution was added and centrifuged at 14,000×g for 40 min at 18 °C, repeated twice; ④ 25 mmol/L $NH_4HCO_3$ solution was added and centrifuged at 14,000×g for 40 min at 18 °C, repeated twice; and ⑤ trypsin (Trypsin Gold, Promega, Trypchburg, WI, USA) was added at a ratio of 1:50 trypsin:protein for digestion and kept in a water bath overnight at 37 °C. The following day, peptides were collected by centrifugation at 13,000×g for 30 min at 4 °C, desalted through an HLB column (Waters, Milford, MA, USA), dried using a vacuum dryer, and stored at −80 °C.

## LC –MS/MS analysis

The digested samples were reconstituted with 0.1% formic acid, and peptides were quantified using a BCA kit, followed by diluting of the peptide concentration to 0.5 µg/ µL. Mixed peptide samples were prepared from 4 µL of each sample and separated using a high pH reversed-phase peptide separation kit (Thermo Fisher Scientific, Waltham, MA, USA) according to the instructions. Ten effluents (fractions) were collected by centrifugation, dried using a vacuum dryer and reconstituted with 0.1% formic acid. iRT reagent (Biognosys, Switzerland) was added at a volume ratio of sample:iRT of 10:1 to calibrate the retention times of extracted peptide peaks. For analysis, 1 µg of each peptide from an individual sample was loaded onto a trap column and separated on a reverse-phase C18 column (50 µm ×150 mm, 2 µm) using the EASY-nLC1200 HPLC system (Thermo Fisher Scientific, Waltham, MA, USA) (*Wu, Guo & Gao, 2017*). The elution for the analytical column lasted 120 min with a gradient of 5%–28% buffer B (0.1% formic acid in 80% acetonitrile; flow rate 0.3 µL/min). Peptides were analysed with an Orbitrap Fusion Lumos Tribrid Mass Spectrometer (Thermo Fisher Scientific, Waltham, MA, USA).

To generate the spectrum library, 10 isolated fractions were subjected to mass spectrometry in data-dependent acquisition (DDA) mode. Mass spectrometry data were collected in high sensitivity mode. A complete mass spectrometric scan was obtained in the 350–1500 m/z range with a resolution set at 60,000. Individual samples were analysed

using Data Independent Acquisition (DIA) mode. DIA acquisition was performed using a DIA method with 36 windows. After every 10 samples, a single DIA analysis of the pooled peptides was performed as a quality control.

## Database searching and label-free quantitation

Data were collected as previously described by *Wei et al. (2019)*. Specifically, raw data collected from liquid chromatography–mass spectrometry were imported into Proteome Discoverer (version 2.1, Thermo Fisher Scientific, Waltham, MA, USA) and the Swiss-Prot rat database (published in May 2019, containing 8,086 sequences) for alignment, and iRT sequences were added to the rat database. Then, the search results were imported into Spectronaut Pulsar (Biognosys AG, Switzerland) for processing and analysis. Peptide abundances were calculated by summing the peak areas of the respective fragment ions in $MS_2$. Protein intensities were summed from their respective peptide abundances to calculate protein abundances.

## Statistical analysis

Two technical replicates were performed for each sample, and the average was used for statistical analysis. In this experiment, the experimental group samples at different time periods were compared before and after, and the control group was set up to rule out differences in growth and development. The identified proteins were compared to screen for differential proteins. The differential protein screening conditions were as follows: fold change (FC) $\geq$ 1.5 or $\leq$ 0.67 between groups and $P$ value $< 0.05$ by two-tailed unpaired t test analysis. The Wu Kong platform was used for the selected differential proteins (https://www.omicsolution.com/wkomics/main/); the UniProt website (Release 2023_01) (https://www.uniprot.org/) and the DAVID database (*Huang, Sherman & Lempicki, 2009*) (https://david.ncifcrf.gov/) were used to perform functional enrichment analysis. In the PubMed database (https://pubmed.ncbi.nlm.nih.gov), the reported literature was searched to perform functional analysis of differential proteins.

# RESULTS

## Characterization of e-cigarette smoking rats

In this experiment, the rats were observed behaviourally during the modelling process. Among them, rats in the control group had normal activity and normal dietary and drinking water intake. Water intake was significantly increased in the treated group compared with the control group. At the same time, the body weight of the rats was recorded every 5 days in this experiment (Fig. 2), and a significant increase in individualized variance in the body weight of the rats in the experimental group was observed.

## Urinary proteome changes
### Urine protein identification

Fifty-five urine protein samples were analysed by LC −MS/MS tandem mass spectrometry after the rat e-cigarette model was established. In total, 1093 proteins were identified ($\geq$ 2 specific peptides and FDR $< 1\%$ at the protein level).

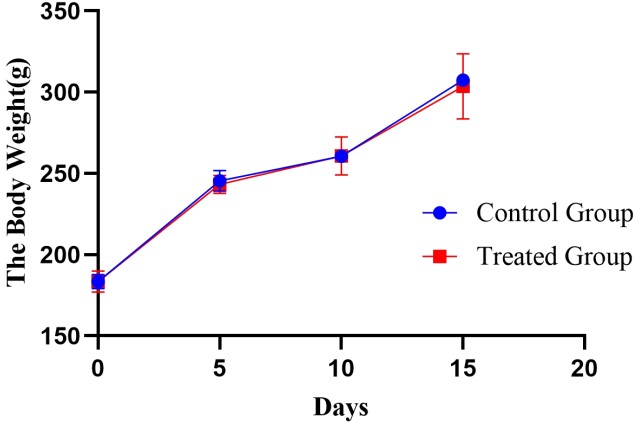

**Figure 2** **Body weight changes in the rat e-cigarette model.** The obtained results are shown as the means ± SDs for the control group ($n = 5$) and the treated group ($n = 6$).

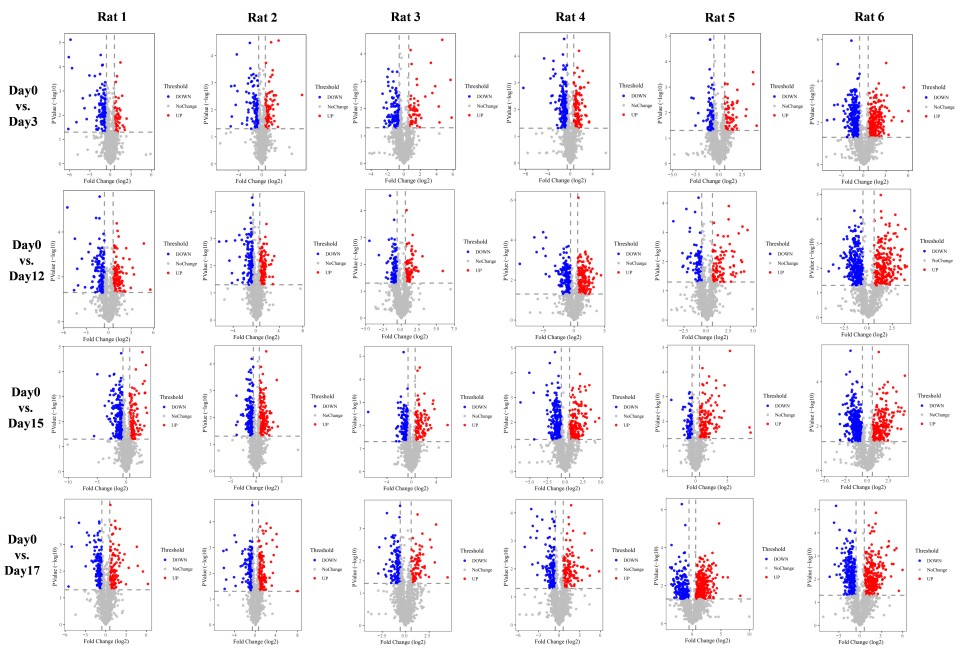

**Figure 3** **Volcano plots of differential proteins produced by six experimental rats at different time points *vs.* Day0.**

### Urine proteome changes

With the aim of investigating whether the changes were consistent in the six treated rats, we performed urine proteomics analysis of each rat individually as its own control and compared them at different time points to D0 (Fig. 3). The screening differential protein

**Table 1  Changes in differential protein expression during e-cigarette smoking in individual rats.**

| Days | Differential protein expression | Control group (pcs) | | | | | Treated group (pcs) | | | | | |
|---|---|---|---|---|---|---|---|---|---|---|---|---|
| | | Rat1 | Rat2 | Rat3 | Rat4 | Rat5 | Rat1 | Rat2 | Rat3 | Rat4 | Rat5 | Rat6 |
| | Total | 215 | 208 | 208 | 162 | 96 | 181 | 203 | 177 | 331 | 144 | 490 |
| Day3/Day0 ↑ | | 89 | 83 | 82 | 71 | 59 | 51 | 87 | 58 | 127 | 60 | 199 |
| ↓ | | 126 | 125 | 126 | 91 | 37 | 130 | 116 | 119 | 204 | 84 | 291 |
| | Total | 401 | 257 | 214 | 275 | 217 | 260 | 265 | 188 | 336 | 173 | 533 |
| Day12/Day0 ↑ | | 175 | 115 | 82 | 126 | 106 | 117 | 112 | 80 | 152 | 93 | 241 |
| ↓ | | 226 | 142 | 132 | 149 | 111 | 143 | 153 | 108 | 184 | 80 | 292 |
| | Total | 288 | 255 | 337 | 289 | 505 | 350 | 341 | 261 | 402 | 246 | 512 |
| Day15/Day0 ↑ | | 124 | 141 | 130 | 104 | 312 | 125 | 153 | 104 | 163 | 151 | 214 |
| ↓ | | 173 | 114 | 207 | 185 | 193 | 225 | 188 | 157 | 239 | 95 | 298 |
| | Total | 395 | 264 | 289 | 320 | 191 | 266 | 283 | 199 | 290 | 475 | 518 |
| Day17/Day0 ↑ | | 183 | 134 | 111 | 157 | 142 | 117 | 140 | 81 | 124 | 312 | 251 |
| ↓ | | 212 | 130 | 178 | 163 | 49 | 149 | 143 | 188 | 166 | 163 | 267 |

conditions were FC $\geq$ 1.5 or $\leq$ 0.67 and two-tailed unpaired t test $P < 0.05$. The differential protein screening results are shown in Table 1.

To observe the consistency of the changes in the six treated rats more intuitively, we made Venn diagrams of the differential proteins screened by comparing the six rats in the experimental group before (D0) and after on D3, D12, D15, and D17 (Fig. 4).

To verify the correlation between rat differential proteins at different time points, we made Venn diagrams. These diagrams show the common differential proteins identified in urine protein from five or more treated rats on D3, D12, D15, and D17 compared with D0 (Fig. 5). Among them, cadherin was screened by 5 or more rats at all four time points, and all showed a tendency towards downregulation. Six differential proteins were screened by 5 or more rats at all three time points, showing more common differences. The trends of common differential proteins in the six rats of the treated group on different days are shown in Table 2.

We analysed the differential proteins produced by D3 after e-cigarette smoking compared with D0 (self-controls) in six treated rats. We found that two differential proteins were commonly identified in six treated rats, and 16 differential proteins were commonly identified in five experimental rats. After comparing the differential proteins produced by the control group before and after with itself, the repeated proteins were screened out, resulting in differential proteins with specific commonalities in the treated group. The details are presented in Table S1.

We also analysed the consensus differential proteins produced by six rats in the experimental group on D12 after e-cigarette smoking versus D0 (self-control). Nine differential proteins were commonly identified in six treated rats, and 18 differential proteins were commonly identified in five experimental rats. After comparison with the differential proteins produced by a single rat in the control group before and after

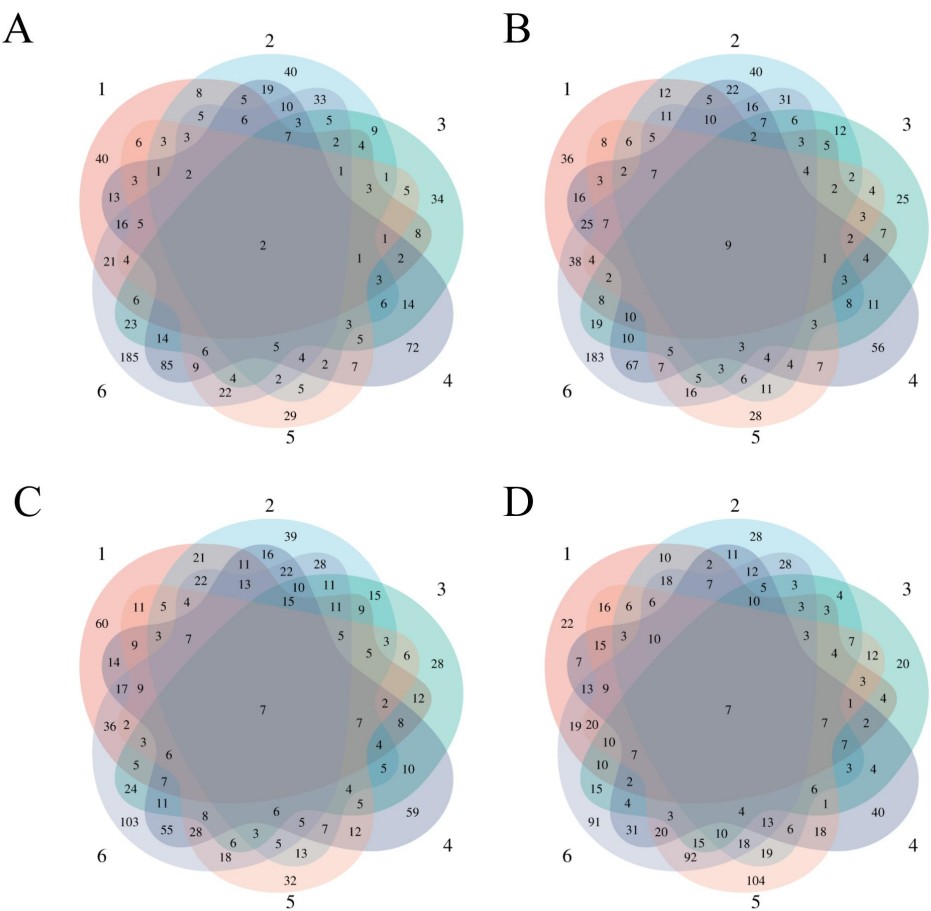

**Figure 4  Differential proteins Venn diagram produced by six treated rats and their own control.** (A) D3 *vs*. D0. (B) D12 *vs*. D0. (C) D15 *vs*. D0. (D) D17 *vs*. D0.

treatment, the repeated proteins were screened out, resulting in differential proteins with specific commonalities in the treated group. The details are presented in Table S2.

When D15 was compared with D0, seven differential proteins were commonly identified in six treated rats, and 45 differential proteins were commonly identified in five experimental rats. After comparing the differential proteins produced by the control group before and after, the repeated proteins were screened out, resulting in differential proteins with specific commonalities in the experimental group. The details are presented in Table S3.

Finally, we analysed six rats in the experimental group for differential proteins screened by self-comparison between D17 and D0, of which seven differential proteins were commonly identified from six rats in the experimental group and 42 differential proteins were commonly identified from five rats in the experimental group. After comparing the differential proteins produced by the control group before and after, the repeated proteins were screened out, resulting in differential proteins with specific commonalities in the
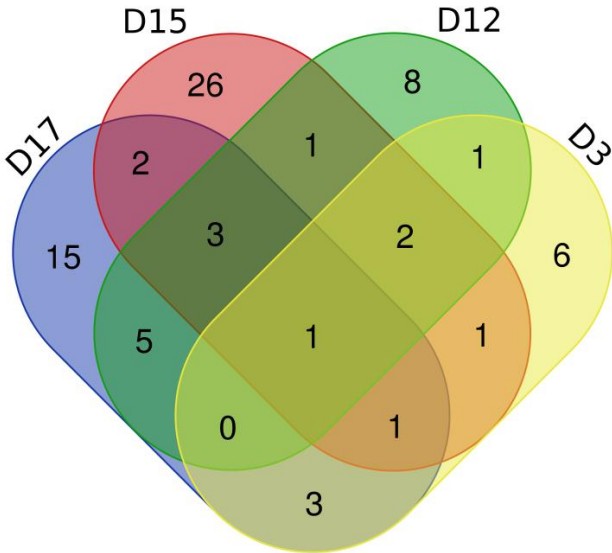

**Figure 5** Venn diagram of common differential proteins in six rats in the experimental group on different days.

experimental group (see Table S4 for details). Among the results presented at different time points, we found some consistency in the effects caused by e-cigarette smoking in rats.

### Functional comparison analysis

To investigate the function of these differential proteins, we performed functional analysis of biological pathways using the DAVID database on the differential proteins coselected from five or more treated rats (see Table S5 for details). Thirty-two of these biological processes were enriched by differential proteins at both time points. We also performed signalling pathway analysis of differentially identified proteins coidentified in five or more treated rats (Table 3). Two of these signalling pathways were coenriched at both time points. These were Legionellosis and Ferroptosis. In addition, we also identified many enriched signalling pathways associated with respiratory diseases, such as the apelin signalling pathway (*Tatemoto et al., 1998*; *Yan et al., 2020*), folate biosynthesis pathway (*Stanisławska-Sachadyn et al., 2019*), and arachidonic acid metabolism (*Giudetti & Cagnazzo, 2012*). Of note, we also found two signalling pathways associated with chemical carcinogenesis, that is, chemical carcinogenesis-DNA adducts and chemical carcinogenesis-reactive oxygen species. These findings reinforce the sensitivity of the urine proteome.

## DISCUSSION

In this study, we constructed a rat e-cigarette model and collected urine samples before, during, and after e-cigarette smoking in rats on days 0, 3, 12, 15, and 17 to explore e-cigarettes from the perspective of urine proteomics. To exclude the influence of individual

**Table 2  Trends of common differential proteins in six rats in the experimental group on different days.**

| UniProt accession | Human ortholog | Protein name | Rat amount (pcs) and trend | | | | Related to OBP | Related to human disease |
|---|---|---|---|---|---|---|---|---|
| | | | D3 | D12 | D15 | D17 | | |
| G3V803 | P19022 | Cadherin-2, Neural cadherin | 5 ↓ | 6 ↓ | 5 ↓ | 5 ↓ | – | *László & Lele (2022)* |
| P14668 | P08758 | Annexin A5 | – | 5 ↑ | 5 ↑ | 5 ↑1↓ | – | *Zhang, Meng & Jing (2022)* |
| Q99041 | P49221 | Protein-glutamine gamma-glutamyltransferase 4 | – | 1 ↑4↓ | 3 ↑2↓ | 2 ↑4↓ | – | – |
| P20761 | – | Ig gamma-2B chain C region | – | 4 ↑1↓ | 5 ↑ | 4 ↑1↓ | – | – |
| B3EY84 | Q9NY56 | Lipocalin 13, Odorant-binding protein 2A | 5 ↓ | – | 5 ↑1↓ | 5 ↑ | *Bri et al. (2002)* | – |
| P07151 | P61769 | Beta-2-microglobulin | 4 ↑1↓ | 4 ↑1↓ | 6 ↑ | – | – | – |
| Q6IRS6 | Q9UGM5 | Fetuin-B | 5 ↓ | 6 ↓ | 6 ↓ | – | – | *Diao et al. (2016)* |
| P27590 | P07911 | Uromodulin | – | – | 5 ↑ | 5 ↑ | – | – |
| Q9JJH9 | – | Alpha-2u globulin | – | – | 5 ↑ | 5 ↑ | *Ponmanickam & Archunan (2006)* | – |
| P0DMW0 | P0DMV8 | Heat shock 70 kDa protein 1A | – | 4 ↑1↓ | – | 3 ↑2↓ | – | *Gál et al. (2011), Hulina-Tomašković et al. (2019) and Parseghian, Hobson & Richieri (2016)* |
| M0RDH1 | – | Odorant-binding protein 2B | – | 5 ↑1↓ | – | 5 ↑ | *Lacazette, Gachon & Pitiot (2000)* | – |
| Q68FP1 | P06396 | Gelsolin | – | 5 ↑ | – | 5 ↑1↓ | – | *Nowak et al. (2015)* |
| A0A0G2K230 | Q14574 | Desmocollin 3 | – | 5 ↑ | – | 5 ↑ | – | *Ezzat & Tahoun (2016)* |
| P51635 | P14550 | Aldo-keto reductase family 1 member A1 | – | 5 ↑ | – | 4 ↑1↓ | – | – |
| Q64724 | – | C-CAM4 | 5 ↓ | – | – | 1 ↑4↓ | – | – |
| P05545 | – | Serine protease inhibitor A3K | 5 ↓ | – | – | 5 ↓ | – | – |
| Q07936 | P07355 | Annexin A2 | 4 ↑1↓ | – | – | 2 ↑3↓ | – | *Huang et al. (2022)* |
| D4A9V5 | – | Lysyl oxidase homolog | – | 6 ↓ | 5 ↓ | – | – | – |
| P46413 | P48637 | Glutathione synthetase | 5 ↓ | – | 5 ↓ | – | – | – |
| D4AE68 | – | Guanine nucleotide-binding protein G(q) subunit alpha | 3 ↑2↓ | 5 ↑1↓ | – | – | – | – |
| Q9QYU9 | – | odorant-binding protein 1F | 5 ↓ | – | – | – | *Aragona et al. (1988)* | – |
| Q9JJI3 | – | Alpha-2u globulin | – | – | 6 ↑ | – | *Ponmanickam & Archunan (2006)* | – |

**Table 3    Signal pathways enriched in common differential proteins produced during e-cigarette smoking in five or more treated rats.**

| Pathway | P-value | | | | Related to human disease |
|---|---|---|---|---|---|
| | D3 | D12 | D15 | D17 | |
| GnRH secretion | 4.20E−02 | – | – | – | – |
| Apelin signaling pathway | 8.80E−02 | – | – | – | *Tatemoto et al. (1998)* and *Yan et al. (2020)* |
| Antigen processing and presentation | – | 4.00E−03 | – | – | – |
| Oestrogen signaling pathway | – | 9.40E−03 | – | – | – |
| Human immunodeficiency virus 1 infection | – | 2.70E−02 | – | – | – |
| Legionellosis | – | 6.30E−02 | – | 9.30E−02 | – |
| Longevity regulating pathway - multiple species | – | 6.90E−02 | – | – | – |
| Renin secretion | – | 7.80E−02 | – | – | – |
| Arrhythmogenic right ventricular cardiomyopathy | – | 8.30E−02 | – | – | – |
| Folate biosynthesis | – | – | 1.10E−03 | – | *Stanisławska-Sachadyn et al. (2019)* |
| Metabolism of xenobiotics by cytochrome P450 | – | – | 6.90E−03 | – | – |
| Chemical carcinogenesis - DNA adducts | – | – | 6.90E−03 | – | – |
| Arachidonic acid metabolism | – | – | 8.00E−03 | – | *Giudetti & Cagnazzo (2012)* |
| Metabolic pathways | – | – | 3.80E−02 | – | – |
| Chemical carcinogenesis - reactive oxygen species | – | – | 5.70E−02 | – | – |
| Ferroptosis | – | – | 6.80E−02 | 6.80E−02 | *Yoshida et al. (2019)* |
| Mineral absorption | – | – | – | 4.00E−03 | – |
| Glycolysis/Gluconeogenesis | – | – | – | 5.60E−03 | – |
| Phenylalanine metabolism | – | – | – | 3.00E−02 | – |
| Histidine metabolism | – | – | – | 4.10E−02 | – |
| Lipid and atherosclerosis | – | – | – | 4.70E−02 | – |
| beta-Alanine metabolism | – | – | – | 5.10E−02 | – |
| Tyrosine metabolism | – | – | – | 6.20E−02 | – |

variance, the experiment used a single rat (before and after) for a controlled analysis, while the control group was set up to rule out differences caused by rat growth and development. From the results presented by the Venn diagram (Fig. 4) of the differential proteins screened from the 6 rats in the experimental group on days 3, 12, 15, and 17 compared with themselves on day 0, we found that most of the differential proteins were personalized, indicating that the effects caused by e-cigarettes on rats had strong individualized variance.

We analysed six treated rats on D3 compared with D0; among the resulting differential proteins, fetuin-B was identified in five rats, all of which showed a significant decreasing trend. There were also significant differences in urine protein on D12 and D15. It has been shown that fetuin-B is a biomarker of COPD (*Diao et al., 2016*), reflecting the sensitivity of the urine proteome. Surprisingly, we also observed odourant-binding proteins (OBPs), including OBP1F and OBP2A, in urine protein with significant changes on D3, of which OBP1F is mainly expressed in the nasal glands of rats (*Aragona et al., 1988*), and OBP2A is also mainly transcribed in the nose of humans and rats (*Bri et al., 2002*). This result suggests that in rats, smoking odourants can actually leave traces in the urine proteome.

Currently, the physiological role of OBPs is not fully understood (*Redl & Habeler, 2022*), and perhaps the urine proteome can play a role in exploring the specific mechanism of action of OBPs. Annexin A2 is widely used as a marker for a variety of tumours (*Huang et al., 2022*). In addition, *László & Lele (2022)* showed that neurocadherin is one of the most important cell adhesion molecules during brain development and plays an important role in neuronal formation, neuronal proliferation, differentiation and migration, axonal guidance, synaptogenesis and synaptic maintenance.

We also compared D12 with D0 of six rats in the experimental group. Among the differential proteins produced in all six treated rats, OBP2B also showed a more consistent upregulation. Unlike OBP2A, OBP2B is mainly expressed in reproductive organs and is weakly expressed in organs of the respiratory system, such as the nose and lung (*Lacazette, Gachon & Pitiot, 2000*). In addition, desmocollin 3 is an essential protein for cell adhesion and desmosome formation and may enhance angiogenesis with metastasis in nasopharyngeal carcinoma and is considered a biomarker for some cancers, such as non-small cell lung cancer (*Ezzat & Tahoun, 2016*). It has also been shown that Annexin A5 may affect the occurrence and development of pathological phenomena, such as tumour diseases, pulmonary fibrosis and lung injury. Annexin A5 may also be used as a biomarker in the study of diseases, such as tumours and asthma, and may promote the occurrence and development of laryngeal cancer and nasopharyngeal carcinoma (*Zhang, Meng & Jing, 2022*). We also screened an important cellular target of the nicotine metabolite cotinine, gelsolin, which may affect basic tumour transformation and metastasis processes, such as migration and apoptosis, through gelatine (*Nowak et al., 2015*). Heat shock protein, on the other hand, is reported to be a major marker affected by cigarette smoke and is involved in signalling pathways associated with the cell cycle, cell death and inflammation (*Gál et al., 2011*; *Hulina-Tomašković et al., 2019*; *Parseghian, Hobson & Richieri, 2016*).

Comparing D15 with D0, the common differential proteins produced by the six experimental rats included α-2u globulin. *Ponmanickam & Archunan (2006)* showed that α-2u globulin may act as a carrier of hydrophobic odourants in the preputial gland, which plays an important role in producing pheromone-communicating olfactory signals in rats. Therefore, α-2u globulin is likely to be involved in the transmission of olfactory signals in rats.

Compared with those on D0, most of the common differential proteins produced by the six rats in the experimental group on D17 were similar to those produced on other days, and all time points contained odourant-binding proteins and proteins associated with a variety of diseases (Table 2).

We performed signalling pathway analysis of differentially expressed proteins coidentified in five or more treated rats (Table 3) and focused on two signalling pathways coenriched by the two time points: Legionellosis and Ferroptosis. Pneumonia caused by Legionnaires' disease may cause damage to the body similar to the mechanism of the effects of smoking e-cigarettes on the body. *Yoshida et al. (2019)* showed that smoking can induce ferroptosis in epithelial cells, and this signalling pathway is involved in the pathogenesis of chronic obstructive pulmonary disease. In addition, we also found enrichment of many signalling pathways associated with respiratory diseases, such as the apelin signalling

pathway (*Tatemoto et al., 1998*; *Yan et al., 2020*), folate biosynthesis pathway (*Stanisławska-Sachadyn et al., 2019*), and arachidonic acid metabolism (*Giudetti & Cagnazzo, 2012*). Apelin is an endogenous ligand for the G protein-coupled receptor APJ (*Tatemoto et al., 1998*), and the apelin/APJ pathway is closely related to the development of respiratory diseases. Targeting the apelin/APJ system may be an effective therapeutic approach for respiratory diseases (*Yan et al., 2020*). A study by *Stanisławska-Sachadyn et al. (2019)* showed that serum folate concentrations were higher in smokers than in healthy controls, and it was postulated that folate synthesis is associated with an increased risk of lung cancer. Because the 23 enriched signalling pathways include multiple signalling pathways directly related to the immune system, cardiomyopathy, and atherosclerosis, we speculated that smoking e-cigarettes may affect the immune system and cardiovascular system of rats. For the two enriched signalling pathways to be associated with chemical carcinogenesis, it may be possible to verify previous findings that e-cigarette smoke contains carcinogenic chemicals (*Eshraghian & Al-Delaimy, 2021*).

There are some limitations in this study. During urine sample processing, we heated protein solutions containing high concentrations of urea (>1 M) at high temperatures (97 °C) to randomly introduce the modification of carbamoylation. Although we used analytical software to exclude this modification, analyzed peptides that did not contain the carbamoylation modification, and this randomly introduced modification did not affect the non-randomized results, other potential modifications may still be present. In the future, we will treat urine samples under milder conditions to exclude this potential effect as much as possible.

## CONCLUSION

There were strong individual variances in the differential proteins produced by rats after smoking e-cigarettes under the same conditions. Fetuin-B, a biomarker of COPD, and annexin A2, which is recognized as a multiple tumour marker, were coidentified in five out of six treated rats' self-control samples on D3. Odourant-binding proteins expressed in the olfactory epithelium were also identified in the urine proteome at multiple time points and were significantly upregulated. How odourant-binding proteins expressed in the olfactory epithelium end up in the urine after smoking e-cigarettes remains to be elucidated. Pathways enriched by the differential proteins include the apelin signalling pathway, folate biosynthesis pathway, arachidonic acid metabolism, chemical carcinogenesis-DNA adducts and chemical carcinogenesis-reactive oxygen species. They have been reported to be associated with immune system, cardiovascular system, respiratory system, etc. Urinary proteome could reflect the effects of e-cigarettes in rats.

### Funding

This research was funded by the National Key Research and Development Program of China (2018YFC0910202), the Fundamental Research Funds for the Central Universities

(2020KJZX002) and the Beijing Natural Science Foundation (7172076). The funders had no role in study design, data collection and analysis, decision to publish, or preparation of the manuscript.

## Grant Disclosures
The following grant information was disclosed by the authors:
The National Key Research and Development Program of China: 2018YFC0910202.
The Fundamental Research Funds for the Central Universities: 2020KJZX002.
The Beijing Natural Science Foundation: 7172076.

## Competing Interests
The authors declare that there are no competing interests.

## Author Contributions
- Yuqing Liu conceived and designed the experiments, performed the experiments, analyzed the data, prepared figures and/or tables, authored or reviewed drafts of the article, and approved the final draft.
- Ziyun Shen performed the experiments, authored or reviewed drafts of the article, and approved the final draft.
- Chenyang Zhao analyzed the data, prepared figures and/or tables, and approved the final draft.
- Youhe Gao conceived and designed the experiments, authored or reviewed drafts of the article, and approved the final draft.

## Animal Ethics
The following information was supplied relating to ethical approvals (i.e., approving body and any reference numbers):

All experimental procedures followed the review and approval of the Ethics Committee of the College of Life Sciences, Beijing Normal University. (Approval No. CLS-AWEC-B-2022-003).

## Data Deposition
The mass spectrometry proteomics data are available at the ProteomeXchange Consortium via the iProX partner repository: IPX0005476000.

https://www.iprox.cn/page/project.html?id=IPX0005476000.

## Supplemental Information
Supplemental information for this article can be found online at http://dx.doi.org/10.7717/peerj.16041#supplemental-information.

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
