# Peer review of "Urine proteomic analysis of the rat e-cigarette model"

_PeerJ, doi:10.7717/peerj.16041_

## Round 0.1 · original submission · Major Revisions

Please address all the issues pointed out by the reviewers and amend the manuscript accordingly.

Reviewer 1 ·

Basic reporting

Fail. (see Additional comments)

Experimental design

Pass.

Validity of the findings

Pass. (See Additional comments)

Additional comments

Dear Editor,

I have reviewed the manuscript entitled "Urine proteomic analysis of the rat e-cigarette model" submitted by Yuqing Liu et al.
This manuscript describes a mass spectrometry-based proteomic profiling study for identification and label-free quantitation of urine proteins as potential biomarkers to evaluate e-cigarette toxicity in rats. As a result, they report an overall identification of 1093 proteins in 55 urine protein samples from 6 e-cigarette treated rats and 5 healthy control rats. Among these proteins, a considerable number of differentially expressed proteins are present, however, with strong individual variation. The authors then highlight and discuss the biological/pathological significance of several proteins of interest such as fetuin-B, annexin A2, and odorant-binding proteins. The authors also suggest, from their pathway analysis result, smoking e-cigarettes may impose negative effects on a variety of systems of the body, including immune, cardiovascular, and respiratory systems.
Overall, the authors have conducted a respectable proteomic profiling experiment for exploring e-cigarette toxicity biomarkers in their rat model. The manuscript is well-written, informative, and presents a scientific deliverable that is potentially of high interest in the biomarker research community.
As a reviewer, I appreciate the authors’ effort and work put into this proteomic profiling study. However, I would like to kindly suggest a few improvements that could strengthen the study and make the conclusions more reliable. I have outlined these points below, and I recommend the authors address these issues before the manuscript can be considered for publication on PeerJ.

Major Concerns:
1. It has come to my attention that the urine proteomic profiling results of the control group, which consisted of 5 healthy rats, were not included, and discussed in this manuscript. It appears to me that the authors only focused on comparing the experimental group at various time points (day 3, 12, 15, and 17) to day 0. By incorporating control group data (part of the 1093 proteins identified, I suppose, as stated in Line 203) and discussion, the authors will be able to provide a more comprehensive understanding of the factors influencing protein differential expression. To enhance this manuscript, I kindly suggest the following:
a. Report the qualitative proteomic profile comparison of experimental group vs. control groups. Such data can be visualized on a Venn Diagram.
b. Report the quantitative proteomic profile comparison experimental group vs. control groups. Such data can be visualized on a Volcano plot.
c. Report individual trending proteins of interest (significantly upregulated or downregulated) with their %abundance (instead of fold change) in experimental group vs. control group at day 0, 3, 6, 9, 12, 15, and 17. Such data can be visualized on a barchart or heatmap.

2. In the “2.3 Treatment of urine samples.” section (Line 118 to Line 148) of the manuscript, the authors describe that each isolated urine protein pellet is reconstituted in a lysis solution that contains 8M urea. The authors then state that the protein reduction is performed as 0.5 μg/μL solution after dilution with 25mM ammonium bicarbonate and addition of 20mM DTT solution. The reaction mixture is then heated to 97 °C and incubated for 5 min. It is not clear that whether the authors heat the protein solution with a high concentration of urea (>1M). If so, the authors may need to check for urea carbamylation on the subsequent urine proteins and tryptic peptides. Urea carbamylation is well known to introduce random artifacts and significantly compromise the overall proteomic profiling results.

Minor Concerns:
1. Line 71: “biomarkers can observe” can be rephrased as ““biomarkers may indicate”.
2. Line 76: Alzheimer's disease and Parkinson's disease are neurodegenerative diseases, which are fundamentally different from psychiatric diseases.
3. Line 90: acronym “SPF” is not defined anywhere in the manuscript.
4. Line 105: the term “3% nicotine smoke bullets” is confusing. Did the authors mean “3% nicotine e-cigarette refill cartridges”?
5. Line 184: The screening criteria (FC) ≥ 1.5 or ≤ 0.67 appear to be arbitrary. The authors may want to add a reference or additional data to justify such settings.
6. Line 203: The authors may want to disclose a detailed list of all 1093 rat urine proteins identified in this study as part of the supplemental information. This will strengthen the credibility of the manuscript.
7. Table 1: No unit for the numeric values on the table. Are they numbers of differentially expressed proteins?
8. Table 2: No unit for the numeric values on the table. Are they fold change (vs D0) or normalized intensity?
9. Table 3: Why are there blank cells on the table? What does it mean if, for instance, GnRH secretion has blank P-value for D12, D15, and D17? Does it mean GnRH secretion pathway is not enriched in those samples or something else?
10. Figure 1 and Figure 2: The authors may consider trimming some white space around the actual graphic scheme.
11. Figure 3: numeric labels around Venn Diagrams are confusing. For instance, in Venn Diagram (A) D3 vs. D0, it’s difficult to understand which, among rat 1 ,2 ,3 ,4 ,5 ,6, is D3 and which is D0.
12. Line 205 to Line 246, also for Figure 3 and 4: The authors may consider performing quantitative analyses of these differential expression results and visualized them on volcano plots.

In conclusion, I believe that the manuscript has the potential to be a valuable addition to PeerJ, but it requires some revisions to address the concerns mentioned above. I recommend that the authors address these concerns and resubmit their manuscript for further consideration.

Sincerely,
Reviewer

Cite this review as

Reviewer 2 ·

Basic reporting

The English language should be improved to ensure that an international audience can clearly understand your text. Some areas where the language could be improved include lines 15-16. I have also annotated some parts of the PDF for correction. The authors should have a colleague who is proficient in English and familiar with the subject matter review their manuscript, or contact a professional editing service.

Abstract:
Lines 15-16 correction: Urine samples were collected from a rat e-cigarette model before, during, and after e-cigarette smoking.

Lines 22-25: The pathways being referenced are too broad. Authors need to granularize their analysis to specify what aspects of the pathways are affected.

Lines 60-63: The claim that glomerular filtration is not regulated by homeostatic mechanisms is not correct. These statements need to be better worded or deleted.

Experimental design

Although the research question is well defined, the investigation conducted is not rigorous. Importantly, the methodology is not described enough in sufficient details to enable replication.

Lines 102-111: The description of the rat model is not clear and adequate. Specifically, how were the made to smoke the e-cigarettes? What brand of e-cigarettes? What apparatus was used for creating the smokes? What is the duration of smoking, were the rats made to smoke for 1h per day, for a total of 14 days? The authors stated that the rats were observed for behavioral changes during the experiments. What type of behaviors were observed and how were these behaviors measured?

Lines 113-117: Which metabolic cages were used for urine collection? Please, specify the model and manufacturer. The way the days of smoking were counted is extremely unclear.

Lines 119-122: Clarifications about how the nonsmoking day 0, smoking days 3 and 12, and smoking cessation days 1 and 3 were arrived at; as well as how these time points fit in the overall timeline of the smoking experiments need to be included. These time points should be conspicuously marked in Figure 1 with different colors to improve visualization. I commend the authors for clearly stating the rationale behind selecting nonsmoking day 0, smoking days 3 and 12, and smoking cessation days 1 and 3. But, beyond this, the authors should also clearly state what they expected or hypothesized to observe from the urine proteomics of these focused time points.

Line 123: What five time points are the authors referring to? Are these five time points the nonsmoking day 0, smoking days 3 and 12, and smoking cessation days 1 and 3 or different time points? This should be clearly stated.

Line 125: The authors utilized ethanol precipitation for urine protein recovery. How did the author ensured the absolute (100% ?) ethanol used did not denature the protein?

Line 184-186: I will be great if the authors could clarify why they have chosen this particular fold change. It is particularly problematic that the Wukong platform (https://www.omicsolution.org/wkomic/main/) which the authors claimed to have used for differential analysis was not available online as at the time of writing this comment. Although references were made to the Uniprot and DAVID websites, the version numbers used for the analysis should be clearly stated.

Validity of the findings

Some of the results obtained were thinly sliced, and visualized in ways that are not meaningful. For instance, figures 3 and 4 are both Venn diagrams filled with numbers, the specific identities of these proteins are not known. These 2 figures could have been easily visualized as one figure.

Because a proteomic analysis involves multiple correction procedure that filters results based on the reality of theoretical versus experimental spectral data, the chances of spurious proteins (false positives) is appreciable. Therefore, statistical procedures need to be combined with downstream validation. The authors did not validate the key proteins observed in their MS/MS analysis. This creates doubts as to the relevance of these proteins to e-cigarette smoking on one hand, and whether these findings could be extrapolated to the human body as they tried to do.

Annotated reviews are not available for download in order to protect the identity of reviewers who chose to remain anonymous.
Cite this review as

---

## Round 0.2 · Major Revisions

As you can see, the reviewers still have some reservations and require additional clarifications.

Reviewer 1 ·

Basic reporting

The authors have made significant efforts to improve the manuscript following the initial review. They have addressed language issues effectively and this is reflected in the quality of the text. The reported use of professional editing services is commendable.

Experimental design

Upon re-review, it is clear that the authors have taken into consideration the concerns regarding the experimental design. They have provided more detailed explanations of the rat e-cigarette model and the proteomic analysis methodology, which are now clearly outlined and justified.

Validity of the findings

The authors have greatly improved the statistical analysis and the discussion sections of the paper. However, there is one point of concern that has not been adequately addressed, which potentially impacts the validity of the findings.
In the process of urine sample treatment, the proteins were heated in a solution with a high concentration of urea. This process could lead to protein carbamylation by urea, which is known to introduce random artifacts and might compromise the overall proteomic profiling results. While the authors have explained that they used software to exclude the spectra of the modified peptides, the potential for error remains, as this process might not account for all possible modifications or artifacts.
I would suggest the authors include a discussion of this potential limitation in the Discussion section of the paper. This would not only provide full transparency to the readers but also set the findings in the appropriate context, acknowledging the potential for error due to the carbamylation of proteins.
By addressing this concern explicitly in the paper, the authors can contribute to the larger scientific discourse on the implications of sample treatment methods on the results of proteomic analyses.

Additional comments

Overall, the authors have shown an exemplary commitment to refining the manuscript based on the initial review comments. The improvements made in the methodology, data analysis, and discussion provide a clearer, more comprehensive view of their work. The figures and tables are now effectively presented, and the findings are well-contextualized within the broader field.

Cite this review as

Reviewer 2 ·

Basic reporting

No comment

Experimental design

No comment

Validity of the findings

New evidence has been added by the authors. The heat maps (Figures 1 to 5) do not reveal distinct clustering of the test (T) versus control (C) groups, leading to a lack of structure. Something seems to be wrong. Can the authors clarify this?

The absence of clustering within these heatmaps makes a validation experiment more important.

Cite this review as

---

## Round 0.3 · Major Revisions

Please address remaining concerns of the reviewer and revise manuscript accordingly.

Reviewer 1 ·

Basic reporting

OK

Experimental design

OK

Validity of the findings

Dear Authors,

Thank you for your response to my initial concern about the potential effects of protein carbamylation on your study's findings. While I understand your approach of focusing solely on non-modified peptides, I must reiterate my concerns about the potential implications of urea-induced carbamylation.

The process of carbamylation is random and can, therefore, affect peptides differently across various samples, even if you use the same method for all samples. As your study employs label-free quantitation (LFQ) methods, these random modifications could lead to variations in LFQ intensity measurements, potentially skewing your results.

The statement in your manuscript, "Peptide abundances were calculated by summing the peak areas of the respective fragment ions in MS2", leads me to infer that sample peptides from the same protein in different samples could end up with different LFQ intensity measurements due to the aforementioned carbamylation.

Given the potential implications of these carbamylation-induced variations, I strongly recommend a more detailed investigation of the data concerning the proteins of specific interest in this study, such as Fetuin-B and annexin A2. A manual inspection of the LFQ results, sequence coverage, and PSM quality of these proteins would be beneficial. This would not only enhance the transparency and credibility of your study but also provide a more nuanced understanding of these particular proteins' roles.

Overall, acknowledging and discussing the potential challenges related to protein carbamylation in the manuscript will reinforce the scientific discourse on proteomic analyses, increase the robustness of your findings, and provide much-needed transparency for the readers.

Looking forward to seeing these points addressed in your revised manuscript.

Additional comments

N/A

Cite this review as

---

## Round 0.4 · accepted · Accept

All remaining concerns were adequately addressed and your revised manuscript is acceptable now.

Reviewer 1 ·

Basic reporting

No more comments.

Experimental design

No more comments.

Validity of the findings

No more comments.

Additional comments

The authors have appropriately addressed all my concerns. The manuscript is now in good shape and I recommend it for publication.

Cite this review as